# Formalin Inactivation of Virus for Safe Downstream Processing of Routine Stool Parasite Examination during the COVID-19 Pandemic

**DOI:** 10.3390/diagnostics13030466

**Published:** 2023-01-27

**Authors:** Pisith Chinabut, Nuntiya Sawangkla, Suphaluck Wattano, Techit Thavorasak, Weluga Bootsongkorn, Anchalee Tungtrongchitr, Pichet Ruenchit

**Affiliations:** 1Department of Parasitology, Faculty of Medicine Siriraj Hospital, Mahidol University, Bangkok 10700, Thailand; 2Center of Research Excellence in Therapeutic Proteins and Antibody Engineering, Department of Parasitology, Faculty of Medicine Siriraj Hospital, Mahidol University, Bangkok 10700, Thailand; 3Siriraj Integrative Center for Neglected Parasitic Diseases, Department of Parasitology, Faculty of Medicine Siriraj Hospital, Mahidol University, Bangkok 10700, Thailand

**Keywords:** COVID-19, formalin, inactivation, infectious diseases, isotonic formalin, normal saline solution, red blood cell, stool examination, white blood cell

## Abstract

During the COVID-19 pandemic, the parasitology laboratories dealing with fecal samples for the diagnosis of gastrointestinal parasitic infections are confronting the unsaved virus-containing samples. To allow for safe downstream processing of the fecal samples, a protocol for preparing a fecal smear is urgently needed. Formalin was tested with or without isotonic forms for virus inactivation using porcine epidemic diarrhea virus (PEDV) as a representative, as it belongs to the Coronaviridae family. The results revealed complete inactivation activity of 10% formalin and 10% isotonic formalin on coronavirus after 5 min of treatment at room temperature. Both also inhibited *Naegleria fowleri* growth after 5 min of treatment at 37 °C without disruption of the structure. In addition to these key findings, it was also found that isotonic formalin could stabilize both red and white blood cells when used as a solution to prepare fecal smears comparable to the standard method, highlighting its value for use instead of 0.9% normal saline solution for the quantification of blood cells without active virus. The 10% isotonic formalin is useful to safely prepare a fecal smear for the diagnosis of parasites and other infections of the gastrointestinal tract during the COVID-19 pandemic.

## 1. Introduction

Coronavirus disease of 2019 (COVID-19) was characterized as a pandemic in March 2020 by the World Health Organization (WHO) [1]. It is caused by the severe acute respiratory syndrome coronavirus 2 (SARS-CoV-2), a member of the same coronavirus family as the Middle East respiratory syndrome coronavirus (MERS-CoV), porcine epidemic diarrhea virus (PEDV), etc. [2]. As of December 2022, more than 600 million people have been infected with SARS-CoV-2 and nearly seven million infected people have died [3]. The virus spreads mainly between people through small liquid particles secreted from the mouth or nose by coughing, sneezing, breathing, speaking or singing, and human-to-human contact [4].

Fever, dry cough, and dyspnea are the most common manifestations of COVID-19, while diarrhea, nausea, vomiting, and abdominal discomfort are the less common features [5]. In addition to nasal and saliva samples, SARS-CoV-2 has been reported to be isolated from the urine of a patient with severe COVID-19 [6]. Furthermore, the shedding of the virus in feces has been documented. SARS-CoV-2 RNA has been detected in 40% to 85% of fecal samples collected from patients with moderate-to-severe COVID-19 [7,8,9]. The infectious SARS-CoV-2 particle has also been isolated from stool samples from COVID-19 patients [10], and stool samples collected from patients with gastrointestinal manifestations [11]. The shedding of SARS-CoV-2 RNA in feces was found to be correlated with gastrointestinal symptoms [12]. Half of the COVID-19 patients with mild to moderate symptoms shed fecal SARS-CoV-2 RNA within the first week after diagnosis. Thirteen percent of the patients continued to shed viral RNA in their feces 4 months after diagnosis, while nearly 4% shed at 7 months [12]. Interestingly, rectal swabs from eight out of ten pediatric cases of COVID-19 were found to persistently test positive for COVID-19 by real-time reverse transcription—polymerase chain reaction (RT–PCR) even after nasopharyngeal testing was negative [13]. This evidence affirms the potential for fecal–oral or fecal–respiratory transmission and warrants the need for an appropriate handling procedure of fecal specimens collected from suspected or confirmed cases of COVID-19 in order to avoid the possible transmission of SARS-CoV-2 from feces. 

During the COVID-19 pandemic, diagnostic laboratories, including the parasitology laboratory, are confronted with unreliable viral inactivation and safety of clinical specimens. The parasitology laboratory must deal with fecal samples to diagnose gastrointestinal parasitic infections, as well as quantification of red and white blood cells in order to evaluate other infections in the gastrointestinal tract. A simple smear technique or direct wet smear is a basic, conventional, gold standard, and the routine method for the diagnosis of parasitic infections by detecting ova and parasites in stool samples under the microscope [14]. On the basis of this technique, normal saline solution (NSS) containing 0.9% sodium chloride is normally used as a solution to prepare a fecal smear. This concentration showed an osmolarity equal to that of blood plasma, indicating its ability to stabilize red and white blood cells shed in fecal samples [15]. Therefore, it is a good choice and was selected to prepare the fecal smears for enumeration of red and white blood cells. However, this solution cannot inactivate SARS-CoV-2, which can be contaminated in the feces. Therefore, to allow safe downstream processing of fecal samples for laboratory personnel, a new and saved protocol of fecal smear preparation during the COVID-19 pandemic is urgently needed. 

Recently, various detergents and chaotropic reagents were shown to completely inactivate SARS-CoV-2 and other coronaviruses, such as 4% sodium dodecyl sulphate (SDS), 1% sodium deoxycholate (SDC), and trifluoroacetic acid (TFA) in a 1:4 ratio, while 6M guanidinium chloride (GdmCl) and 8M urea partially inactivate [16]. Chlorhexidine digluconate, anionic surfactant, and calcium bicarbonate with a mesoscopic structure (CAC-717) were also found to exhibit virucidal activity against SARS-CoV-2 [17]. Formalin is another chemical agent that is used to inactivate many types of viruses, such as influenza A virus, adenovirus, cytomegalovirus, hepatitis A virus, and poliovirus [18,19,20]. It reacts with amino acids of target viral proteins acting on the N-terminal amino group and side-chains of arginine, cysteine, histidine, and lysine residues to form reversible methylol adducts and nonreversible methylene bridges [21]. However, formalin inactivation activity against coronaviruses and its application to stool examination for the safety of coronavirus transmission through fecal specimens has not been studied extensively. Therefore, the objective of the present study is to evaluate the inactivation activity of formalin against coronavirus, using PEDV as the representative, and its effectiveness in red and white blood cells analyses. The results of the present study will suggest suitable methods for handling SARS-CoV-2-containing fecal specimens. Additionally, it is possible to note that this unexpected material could be handled in a laboratory setting of standard biosafety level-2 (BSL-2).

## 2. Materials and Methods

### 2.1. Human Ethics

Human ethics was reviewed and exempted by the Siriraj Institutional Review Board of the Faculty of Medicine Siriraj Hospital, Mahidol University, Thailand (SIRB Protocol No. 499/2564) because the samples used in this study were not collected directly from patients.

### 2.2. Microorganisms and Cell Line

Porcine epidemic diarrhea virus (PEDV) P70 strain, a coronavirus that does not pose a risk to human health and can be handled in a biosafety level-2 (BSL-2) laboratory, and *Naegleria fowleri* CDC VO 3081 strain, a free-living parasite that can be easily cultured, were used as representatives to test for chemical inactivation by formalin. The African green monkey kidney cell (Vero) was used for the propagation of PEDV. The PEDV and Vero cell line were obtained from Emeritus Professor Dr. Wanpen Chaicumpa at the Center of Research Excellence in Therapeutic Proteins and Antibody Engineering. *N. fowleri* was obtained from Dr. GS Visvesvara at the Centers for Disease Control and Prevention (USCDC).

### 2.3. Chemical Agents

The normal saline solution (NSS) at 0.9% (*w/v*) of sodium chloride (NaCl) was from Thai Nakorn Patana Co., Ltd. 10% formalin was prepared by mixing 100 mL of formaldehyde solution (KEMAUS, Bangkok, Thailand) with 1 L of sterile distilled water. Isotonic formalin (10%; *v/v*) was prepared by dissolving 0.9 g of NaCl in 10% formalin.

### 2.4. Propagation of Porcine Epidemic Diarrhea Virus

PEDV was propagated in African green monkey kidney cell (Vero). Briefly, Vero cells were grown in Dulbecco’s modified Eagle’s medium (DMEM) (Gibco, Life Technologies Corporation, Grand Island, NY, USA) supplemented with 10% heat-inactivated fetal bovine serum (FBS; HyClone, GE Healthcare Bio-Sciences Austria GmbH, Pasching, Austria), 2 mM L-alanyl-L-glutaminase dipeptide (Gibco, Life Technologies Corporation, Grand Island, NY, USA), 100 UmL^−1^ penicillin (Gibco, Life Technologies Corporation, Grand Island, NY, USA), and 100 µgmL^−1^ streptomycin (Gibco, Life Technologies Corporation, Grand Island, NY, USA). The cells were cultured at 37 °C in a humidified incubator with 5% CO_2_. Then, cells were collected and washed twice with sterile phosphate buffered saline (PBS), pH 7.4, and transfected with PEDV at MOI 0.001. The virus was kept at 37 °C in a humidified incubator with 5% CO_2_ until use.

### 2.5. Cultivation of Naegleria fowleri

*N. fowleri* was grown in Nelson’s medium supplemented with 10% FBS at 37 °C in a T 25 cm^2^ flask (Corning, New York, NY, USA) in a secure facility.

### 2.6. In Vitro Inactivation of Coronavirus and Plaque Formation Assay

To test whether a coronavirus could be inactivated with 10% formalin or 10% isotonic formalin, a plaque formation assay was performed using the PEDV P70 strain as the representative [22]. Briefly, 100 µL of 10% formalin or 10% isotonic formalin were mixed with 100 µL of PEDV aliquots (178,000 pfu) and incubated at room temperature for 5 min. After dilution with DMEM at 1:10,000, 1 mL of each of the mixtures were then added to separate wells (triplicate) of 24-well tissue culture plates containing a confluent monolayer of Vero cells and incubated at 37 °C in a humidified incubator with 5% CO_2_ for an hour to allow virus attachment to cells. Virus in medium alone and noninfected Vero cells (mock) were used as negative inactivation control and negative plaque formation control, respectively. The free PEDV particles were removed, and the cells were washed twice with PBS. Carboxymethyl cellulose (CMC) (2% CMC in DMEM supplemented with 2 µgmL^−1^ of TPCK treated trypsin) was added to each well. The plates were incubated at 37 °C in a humidified incubator with 5% CO_2_ for 48 h. Cells were fixed with 10% formalin at room temperature for 1 h before staining with 1% crystal violet dye in 10% ethanol. The inactivation of PEDV was observed by the absence of plaque formation in Vero cells. The number of plaques was counted by light microscopy at 40× magnification. 

### 2.7. Inactivation of Parasite

To evaluate the inactivation activity of 10% formalin and 10% isotonic formalin against parasites, *N. fowleri* (a free-living amoeba) was selected as a representative for testing. Logarithmic phase *N. fowleri* trophozoites (1 × 10^4^ cells) in 100 µL of Nelson’s medium supplemented with 10% FBS, 100 UmL^−1^ penicillin, and 100 µgmL^−1^ streptomycin were seeded in individual wells of 96-well tissue culture plates, and incubated at 37 °C for 24 h. After discarding the culture medium, 100 µL of 10, 5, and 2.5% formalin and isotonic formalin were added to the wells containing the amoebae. The amoebae wells with medium alone served as a negative inactivation control, while the amoebae wells with 0.9% NSS served as a positive inactivation control. The plates were kept at 37 °C for 5 min. The number of viable amoebae in each treatment was determined using the CellTiter-Glo^®^ 3D Cell Viability Assay (Promega, Promega Corporation, Madison, WI, USA). The percentage of trophozoite survival was calculated as follows: [(luminous intensity of the test sample/luminous intensity of the negative inactivation control sample)] × 100. Results are shown as mean ± standard deviation (SD) of a representative of 3 independent experiments.

### 2.8. Preparation of Fecal Smear

Fecal smears were prepared using 3 reagents, 0.9% NSS (a standard solution), 10% formalin, and 10% isotonic formalin (as developed in this study) [23]. Briefly, a drop of stool sample was thoroughly mixed with a drop of 0.9% NSS, 10% formalin, or 10% isotonic formalin on a glass slide, smeared as a thin film, covered with a glass cover slip, and left at room temperature for 5, 10, 20, and 30 min. All experiments were performed under the class II biological safety cabinet in the BSL-2 laboratory in the Department of Parasitology of the Faculty of Medicine Siriraj Hospital, Mahidol university.

### 2.9. Examination of Parasite

The fecal smears prepared with 0.9% NSS, 10% formalin, and 10% isotonic formalin were investigated for ova and parasites under a microscope at 400× magnification.

### 2.10. Enumeration of Red and White Blood Cells

The numbers of red blood cells (RBC) and white blood cells (WBC) presented in each fecal smear prepared above were counted under microscope at 400× magnification, and grouped into the ranges of 0–1, 2–3, 4–5, 6–10, 11–20, 21–30, 31–50, and >50 Cells/High Dry (HD). The counting step was performed by the three independent laboratory technicians. The number of RBC and WBC detected in the smears were shown as percentages. The results of two methods (fecal smears prepared using 0.9% NSS and 10% isotonic formalin) were compared. 

### 2.11. Statistical Analysis

The measurement of agreement between the two methods (fecal smears prepared with 10% isotonic formalin and 0.9% NSS) and inter-rater reliability among the three laboratory technicians were evaluated using IBM SPSS Statistics for Windows, version 21 (IBM Corp., Armonk, NY, USA) and shown as the Kappa coefficient and the Fleiss kappa value in a 95% confidence interval (CI), respectively. The results were considered significantly different at *p*-value < 0.05.

## 3. Results and Discussion

### 3.1. Inactivation Activiy of Formalin against Coronavirus

To test the inactivation activity of formalin to porcine epidemic diarrhea virus, a representative of coronaviruses, PEDV was treated in vitro with 10% formalin and 10% isotonic formalin at room temperature for 5 min. Viability of viruses was evaluated using a plaque formation assay using Vero cells as the host (Figure 1a). PEDV was found to be completely inactivated by 10% formalin and 10% isotonic formalin after 5 min of treatment. Plaque formation was not detected in PEDV treated with 10% formalin or 10% isotonic formalin, while it was observed in untreated PEDV as shown in Figure 1b. This data indicated that PEDV, a species of coronaviruses, could be neutralized with formalin in a short period (5 min). Previously, it has been reported that PEDV was susceptible to 4% anhydrous sodium carbonate, 1% iodophores in phosphoric acid, and 2% sodium hydroxide [24]. Even though 1% formalin has been reported to inactivate PEDV [24], the isotonic form of this agent has not been studied. This study reported for the first time about inactivation activity of isotonic formalin to PEDV. In terms of duration of action, findings of the present study are consistent with previous studies showing that SARS-CoV-2 was completely inactivated by Trizol and Trizol LS (1:4 virus:Trizol reagent) at room temperature for 5 min [25]. Meanwhile, 0.5% SDS, 0.5% Triton X-100, 0.5% NP-40, 100% methanol, and 4% paraformaldehyde were also shown to neutralize SARS-CoV-2 at room temperature, but required a longer duration of treatment (30 min) [25]. Since PEDV and SARS-Cov-2 are viruses in the same family and shared some similar biochemical and biological properties, such as the membrane fusion processes, global folding of M protein, etc. [2,26], sensitivity of these viruses to formalin may be the same. Therefore, 10% formalin and 10% isotonic formalin are good candidates to be used as virus-inactivating agents in laboratories that perform downstream work on samples that contain SARS-CoV-2.

### 3.2. Formalin Inactivation of Free-Living Parasite

To test the ability of formalin to inactivate the parasite, *N. fowleri*, which is a free-living amoeba that causes primary amoebic meningoencephalitis in humans by invading the nostrils through water or dust contaminated with the parasite [27], was tested for in vitro inhibition by formalin and isotonic formalin at different concentrations. Parasite viability was evaluated using the CellTiter-Glo^®^ 3D Cell Viability Assay, which accurately measured cell viability based on the quantitation of adenosine triphosphate (ATP) (Figure 2a) [28]. It was found that all concentrations (10, 5, and 2.5%) of formalin and isotonic formalin completely inactivated the parasite after 5 min of treatment at 37 °C, while the untreated parasite was alive (Figure 2b). This result is in accordance with the previous study, which reported that *Toxoplasma* oocysts were killed by 10% formalin at room temperature, but four days of treatment were required [29]. Similarly, *Leishmania major* promastigotes were also killed by 0.1% formalin, but with an overnight time of treatment required [30]. As expected, 0.9% NSS also inhibited the growth of *N. fowleri,* since it is a well-known low salinity tolerant organism. The growth of *Naegleria* spp. has been reported to be inhibited under 0.2% NaCl conditions [31]. In the present study, it was found that although *N. fowleri* was completely inactivated, its morphology was still intact (data not shown). To confirm this evidence, a stool sample was then sampled to prepare the fecal smears using 10% formalin and 10% isotonic formalin, and the ova and parasites were investigated. The presence of ova and parasites was compared with that found in the smear prepared by the standard method. Based on this test, the vacuolar form of *Blastocytis* spp. found in a fecal smear prepared with 0.9% NSS was also observed in smears prepared with 10% formalin and 10% isotonic formalin. However, its vacuole appears to be more distinctive in a smear prepared with 0.9% NSS and 10% isotonic formalin (Figure 3). Therefore, it could be concluded that, in addition to coronaviruses, formalin also showed inactivation activity against a parasite and still preserved the morphology of the parasite, which will be beneficial for the examination of stool. It is not surprising that formalin maintained the structure of the parasite, because it is a well-known fixative agent used most often for newly obtained samples, including stool samples. It stabilized the biological structure by penetrating the sample, binding to amino acids in all proteins, reacting to uncharged reactive amino groups, and causing cross-links [32]. However, it should be noted that formalin is not suitable for the diagnosis of parasites whose motility is needed for the diagnosis.

### 3.3. Use of Formalin to Prepare Fecal Smears for the Analysis of Red and White Blood Cells

In addition to investigating ova and parasites, the Parasitology laboratory also enumerated red and white blood cells presented in fecal samples to evaluate other infections in the gastrointestinal tract. Since 10% formalin and 10% isotonic formalin showed inactivation activity for both a coronavirus and a parasite, it would be worth if it could also be used to prepare a fecal smear for the enumeration of RBC and WBC to diagnose other infections in the gastrointestinal tract. To test this, 10% formalin and 10% isotonic formalin were applied to prepare fecal smears and tested for stabilization of both RBC and WBC (Figure 4a). The results showed that 10% formalin could not maintain the RBC found in fecal samples (Figure 4b). Lysis of RBC was found when 10% formalin was used to prepare the fecal smears. Contrarily, both RBC and WBC could be stabilized in fecal smears prepared with 10% isotonic formalin or 0.9% NSS, a standard solution, at 0 and 30 min after preparation (Figure 4b). This evidence supports the use of 10% isotonic formalin for the preparation of fecal smears because of its ability to stabilize RBC and WBC, and its inactivation activity against the coronavirus.

To demonstrate the capacity to use 10% isotonic formalin for the quantitative evaluation of RBC and WBC using a fecal smear method, 59 stool samples in which RBC or WBC were detected by the standard method (the fecal smears prepared with 0.9% NSS) were used to prepared fecal smears using 10% isotonic formalin. Both RBC and WBC were found to also be detected in all fecal smears prepared using 10% isotonic formalin. Most fecal smears (27.1 and 25.4%) contain RBC and WBC at more than 50 cells per high dry (cells/HD) (Table 1).

Furthermore, it was found that the numbers of blood cells found in 10% isotonic formalin prepared fecal smears were similar to that found in the 0.9% NSS prepared fecal smears in all defined cell ranges and time intervals (Kappa coefficient value at 1.00) (Table 2). The consistency of the interpretation results between the two methods was also high (100%) at all time intervals (Table 2). These data indicated that 10% isotonic formalin is a good stabilizer of blood cells present in the fecal smears.

To assess the reliability of the agreement of the enumeration of blood cells between the technicians, the three technicians independently performed the enumeration of RBC and WBC in fecal smears prepared with 10% isotonic formalin or 0.9% NSS (Figure 3a). The numbers of positive blood cell samples were not different (Fleiss kappa value 0.918 and 0.932 for RBC and WBC, respectively) (Table 3).

This study confirms the safe use of 10% isotonic formalin for the complete inactivation of the porcine epidemic diarrhea virus in vitro. This agent is likely to be effective against all coronaviruses, especially the SARS-CoV-2, due to their similar biochemical and biological properties. Furthermore, 10% isotonic formalin also inactivated *N. fowleri*, a representative of parasites, without any destruction of their structures. Likewise, it also showed its ability to preserve red and white blood cells, indicating application to RBC and WBC analysis for the diagnosis of other pathogen infections. This evidence may provide guidance for safe laboratory practice in all clinical and research laboratories that handle biological specimens that may contain SARS-CoV-2. However, the formalin hazard remains a concern, even though it can be significantly reduced by using masks and special ventilation systems [33].

## 4. Conclusions

For the preparation of a fecal smear for the diagnosis of parasitic infections and the detection of RBC and WBC for the investigation of other gastrointestinal tract infections during the COVID-19 pandemic, 10% isotonic formalin can be applied instead of 0.9% NSS to ensure the safety of the downstream process in routine stool examination. It not only disinfects coronavirus and the parasite, but also stabilizes the red and white blood cells present in the fecal smears. Therefore, 10% isotonic formalin can be used for the preparation of fecal smears to ensure that laboratory personnel will be safe while working during the COVID-19 pandemic.

## Figures and Tables

**Figure 1 diagnostics-13-00466-f001:**
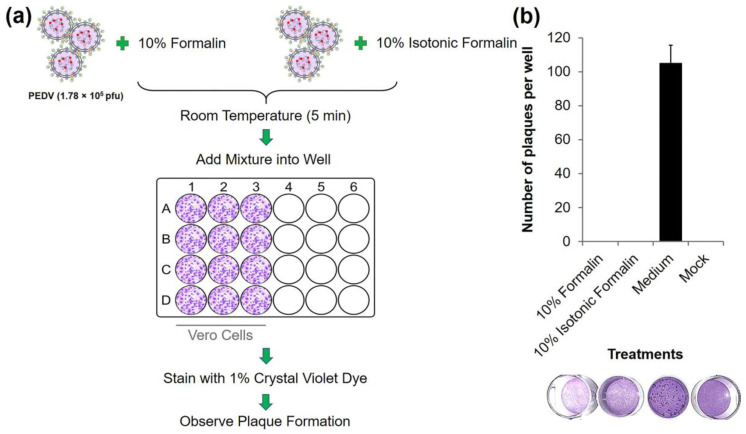
Inactivation activities of 10% formalin and 10% isotonic formalin on porcine epidemic diarrhea virus (PEDV): (**a**) Flow diagram of in vitro inactivation assay. (**b**) Number of plaques per well of 10% formalin- and 10% isotonic formalin-treated PEDV at 5 min of treatment at room temperature. Virus in medium alone and noninfected Vero cells (mock) were used as a negative inactivation control and a negative plaque formation control, respectively. The results are shown as the mean ± standard deviation (SD) of a representative of three independent experiments.

**Figure 2 diagnostics-13-00466-f002:**
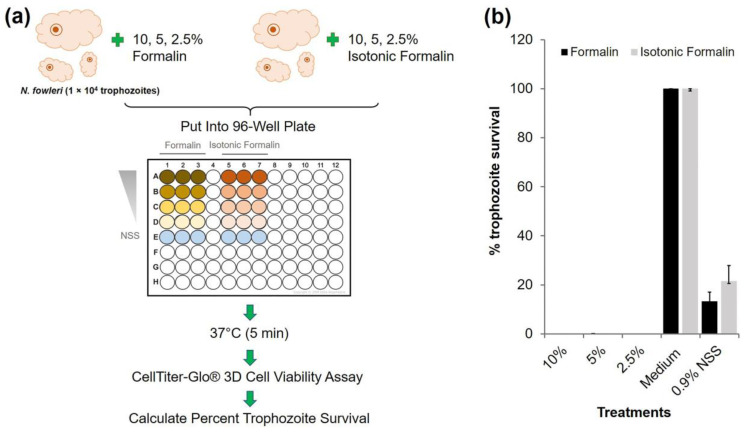
Inactivation activities of formalin and isotonic formalin at different concentrations against *Naegleria fowleri*: (**a**) Flow diagram of inactivation assay. (**b**) Percent survival of *N. fowleri* trophozoites treated with formalin and isotonic formalin 5 min at 37 °C. The results are shown as the mean ± standard deviation (SD) of a representative of three independent experiments.

**Figure 3 diagnostics-13-00466-f003:**
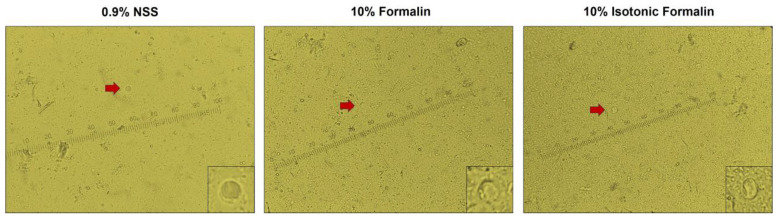
Presence of parasite in the fecal smears. Fecal smears were prepared using different solutions, including 0.9% normal saline solution (NSS) (**left**), 10% formalin (**middle**), and 10% isotonic formalin (**right**). Ova and parasites were observed. One block of scale bar represents 0.25 micrometers. The red arrow indicates the vacuolar form of *Blastocystis* spp. The lower right corner of each figure displays a zoomed-in image of the vacuolar form.

**Figure 4 diagnostics-13-00466-f004:**
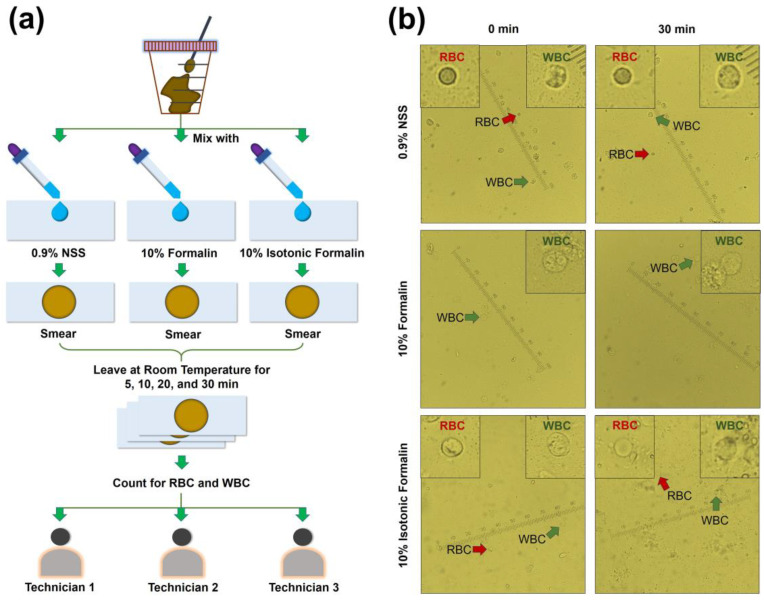
Stabilization of 10% formalin and 10% isotonic formalin blood cells at 0 and 30 min: (**a**) Flow diagram of fecal smear preparation and analysis of blood cells. (**b**) Persistence of red and white blood cells in fecal smears prepared with 10% formalin and 10% isotonic formalin compared to that prepared with 0.9% normal saline solution (NSS), a standard solution. One block of scale bar represents 0.25 micrometers. Red arrow indicates red blood cell (RBC). Green arrow indicates white blood cell (WBC). The upper right and left corners of each figure display zoomed-in images of white and red blood cells, respectively.

**Table 1 diagnostics-13-00466-t001:** Number (Percentage) of the fecal smear samples prepared using 10% isotonic formalin in which RBC and WBC were detected at different numbers (Cells/HD) (n = 59).

Number of Blood Cells (Cells/HD ^1^)	Red Blood Cell Number (Percentage)	White Blood Cell Number (Percentage)
0–1	7 (11.9)	12 (20.3)
2–3	10 (16.9)	11 (18.6)
4–5	7 (11.9)	9 (15.3)
6–10	5 (8.5)	6 (10.2)
11–20	2 (3.4)	4 (6.8)
21–30	11 (18.6)	2 (3.4)
31–50	1 (1.7)	ND ^2^
>50	16 (27.1)	15 (25.4)
Total	59 (100.0)	59 (100.0)

^1^ High Dry. ^2^ Not detected.

**Table 2 diagnostics-13-00466-t002:** Measurement of RBC and WBC agreement between fecal smears prepared with 10% isotonic formalin and standard method (the fecal smears prepared with 0.9% NSS) at different time intervals (n = 59).

Time (Min)	Kappa Coefficient Value	Concordance of Reliable Data	*p*-Value
5	1.00	59 (100%)	<0.001 ^1^
10	1.00	59 (100%)	<0.001 ^1^
20	1.00	59 (100%)	<0.001 ^1^
30	1.00	59 (100%)	<0.001 ^1^

^1^ Significantly different at *p*-value < 0.05.

**Table 3 diagnostics-13-00466-t003:** Reliability of agreement (inter-rater reliability) of blood cell enumeration among the three independent technicians.

Number of Blood Cells	Fleiss Kappa Value (95% CI)	*p*-Value
RBC	0.918 (0.852–0.983)	<0.001 ^1^
WBC	0.932 (0.865–0.998)	<0.001 ^1^

^1^ Significantly different at *p*-value < 0.05.

## Data Availability

Not applicable.

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
