# Peer review of "Formalin Inactivation of Virus for Safe Downstream Processing of Routine Stool Parasite Examination during the COVID-19 Pandemic"

_diagnostics, 2023, doi:10.3390/diagnostics13030466_

Round 1

Reviewer 1 Report

I have reviewed the manuscript entitled “Formalin Inactivation of Virus for Safe Downstream Processing of Routine Stool Parasite Examination During the COVID-19 Pandemic” and found that

The introduction is well-written.

The aim and material & methods are clearly described.

Figures and tables are representable.

In the Results & discussion section, please mentioned some previous similar studies and correlate them with the current one.

There are many syntax errors and grammatical mistakes in the manuscript (for ex. Base should be replaced by based in line 66). Need to rectify them. There is a need to improve the English of the manuscript.

Author Response

Dear Reviewer,

Thank you very much for your brilliant suggestions. Please see the attachment for point-by-point response.

Best Regards,

Pichet Ruenchit, Ph.D.

1
Response to Reviewer 1 Comments
Point 1: In the Results & discussion section, please mentioned some previous similar studies and 
correlate them with the current one.
Response 1: As reviewer’s suggestion, results of the previous studies were added in the Results and 
Discussion section as shown on lines 211-215, 221-224, 246-247, and 248-249. Related references were 
also added in the References section.
Point 2: There are many syntax errors and grammatical mistakes in the manuscript. There is a need 
to improve the English of the manuscript.
Response 2: English grammar was revised throughout the manuscript by native English speaker. 
Certification of proof reading was attached herein. All changes were indicated by track change.
2
Point 3: Base should be replaced by based in line 66. Need to rectify them.
Response 3: “Base on this technique” on line 66 was replaced by “On the basis of this technique” (as 
indicated on line 71 in the revised version).

Reviewer 2 Report

The authors used porcine Epidemic diarrhea virus (PEDV) to test for coronavirus inactivation with or without isosmolar form of formalin and found that isosmolar 10% formalin shows potential as a novel substance to prepare fecal smears for the safe diagnosis of parasites and other gastrointestinal infections during the COVID-19 pandemic. However, it should be noted that 10% formalin has been widely used in studies related to COVID-19 (PMID: 32526193), and whether PEDV is sufficiently representative also needs to be further explained. The authors need to go a step further to clarify the novelty of the manuscript.

Major suggestions that will improve the quality of the manuscript:

1. The title of the manuscript should be revised. In this study, the authors used porcine epidemic diarrhea virus (PEDV) for research, and there was no direct research evidence about COVID-19.

2. The manuscript has always been discussed based on COVID-19, but it uses porcine epidemic diarrhea virus (PEDV) for research. It needs to further explain the similarities and differences between PEDV and COVID-19.

3. The manuscript was not fully discussed. The manuscript has a total of 27 references, 21 of which are in the preface. The discussion should be further deepened and the necessary references cited.

Minor suggestions that will improve the quality of the manuscript:

1. Key words need to be further simplified and accurately expressed (Line 28).

2. This part should be supported by references: 2.6 In Vitro Inactivation of Coronavirus and Plaque Formation Assay (Line 128 ); “2.7. Inactivation of Parasite” (Line 145); “2.8. Preparation of Fecal Smear” (Line 160).

3. The statistical software and significance comparison method used need to be further described (Line 179).

4. The picture is not clear (Fig 3).

5. The format of the reference document is incorrect and the page number is incomplete. It needs to be checked and verified (Line 351).

Author Response

Dear Reviewer,

Thank you very much for your brilliant suggestions. Please see the attachment for point-by-point response.

Best Regards,

Pichet Ruenchit, Ph.D.

1
Response to Reviewer 2 Comments
Point 1: The authors used porcine epidemic diarrhea virus (PEDV) to test for coronavirus inactivation 
with or without isosmolar form of formalin and found that isosmolar 10% formalin shows potential 
as a novel substance to prepare fecal smears for the safe diagnosis of parasites and other 
gastrointestinal infections during the COVID-19 pandemic. However, it should be noted that 10% 
formalin has been widely used in studies related to COVID-19 (PMID: 32526193), and whether PEDV 
is sufficiently representative also needs to be further explained. The authors need to go a step further 
to clarify the novelty of the manuscript.
Response 1: Even though 10% formalin has been used in studies related to SARS-CoV-2, its isotonic 
form has not been studied. The present study investigated for the first time about inactivation activity 
of isotonic formalin to coronavirus. 
As reviewer mentioned about using PEDV as representative of coronaviruses, I think so that 
using SARS-CoV-2 in experiment is the best choice. However, as we mentioned in the manuscript, 
PEDV is a species of coronaviruses that does not pose a risk to human health and can be handled in 
a biosafety level-2 (BSL-2) laboratory which we can accessed. So, we decided to used PEDV instead. 
In our point of view, PEDV is sufficiently representative since it is in the same family with SARSCoV-2 and shared similar biochemical and biological properties, such as the membrane fusion 
processes, global folding of M protein, etc. (indicated by references 2 and 26). To highlight this 
explanation, this sentence was added in the revised version of manuscript (on lines 221-224).
Point 2: The title of the manuscript should be revised. In this study, the authors used porcine 
epidemic diarrhea virus (PEDV) for research, and there was no direct research evidence about 
COVID-19.
Response 2: For this issue, our team have discussed and request to use the original title because even
we used PEDV in this research and there was no any evidence about COVID-19, but we used the 
word “COVID-19” to indicate the situation of “COVID-19 pandemic”. It did not represent the SARSCoV-2 virus.
Point 3: The manuscript has always been discussed based on COVID-19, but it uses porcine epidemic 
diarrhea virus (PEDV) for research. It needs to further explain the similarities and differences 
between PEDV and COVID-19.
Response 3: As reviewer’s suggestion, similarities between PEDV and SARS-CoV-2 were explained 
in the revised version of manuscript (on lines 221-224).
Point 4: The manuscript was not fully discussed. The manuscript has a total of 27 references, 21 of 
which are in the preface. The discussion should be further deepened and the necessary references 
cited.
2
Response 4: As reviewer’s suggestion, results of the previous studies were added in the Results and 
Discussion section to make discussion part more completed (as shown on lines 211-215, 221-224, 246-
247, and 248-249). Related references were also added in the References section.
Point 5: Key words need to be further simplified and accurately expressed (Line 28).
Response 5: A key word “routine stool parasite examination” was replaced by “stool examination”.
Point 6: This part should be supported by references: “2.6 In Vitro Inactivation of Coronavirus and 
Plaque Formation Assay” (Line 128 ); “2.7. Inactivation of Parasite” (Line 145); “2.8. Preparation of 
Fecal Smear” (Line 160).
Response 6: Parts 2.6 and 2.8 of Materials and Methods section were added for related references 
(as shown on lines 141 and 174, respectively). However, citation was not added for part 2.7 since the 
methods described in this section were developed in this study.
Point 7: The statistical software and significance comparison method used need to be further 
described (Line 179).
Response 7: The statistical software and significant comparision method were added in the revised 
manuscript (as shown on lines 196-197 and 199). 
Point 8: The picture is not clear (Fig 3).
Response 8: All four figures were already adjusted for high resolution (300 dpi). Anyway, low 
clearance of the figures caused by PDF inverter. There are some word changes in the figures; so, all 
of them were replaced by the revised versions.
Point 9: The format of the reference document is incorrect and the page number is incomplete. It 
needs to be checked and verified (Line 351).
Response 9: All references were checked and revised.
